# FASTER TRAINING BY SELECTING SAMPLES USING EMBEDDINGS

## ABSTRACT

Long training times have increasingly become a burden for researchers by slowing down the pace of innovation, with some models taking days or weeks to train. In this paper, a new, general technique is presented that aims to speed up the training process by using a thinned-down training dataset. By leveraging autoencoders and the unique properties of embedding spaces, we are able to filter training datasets to include only those samples that matter the most. Through evaluation on a standard CIFAR-10 image classification task, this technique is shown to be effective. With this technique, training times can be reduced with a minimal loss in accuracy. Conversely, given a fixed training time budget, the technique was shown to improve accuracy by over 50%. This technique is a practical tool for achieving better results with large datasets and limited computational budgets.

## 1 INTRODUCTION

As models become more sophisticated and complex, the time required to train a neural network becomes an increasingly important consideration in the development process. Some networks can take a month to train, even on a multi-GPU system (Radford et al., 2017), so any improvement in this training time can result in a faster development cadence.

In this paper, we present a technique that reduces training time through dataset optimization. Our technique involves the analysis of a dataset's embedding space to estimate an optimal training subset. Intuitively, one can imagine that early on in the training process, certain samples can more directly lead to convergence. For example, an object classification network can accomplish more learning by being presented with two very different images than two very similar images of the same object class. Thus, by targeting a network with such distinct samples in its infancy, one can reduce the network's loss more effectively. The problem is now reduced to finding these distinct samples within the training dataset. We accomplish this task using a dense representation that is learned *a priori* in an unsupervised manner using an auto-encoder. By leveraging properties of the embedding space, we are able to find distinct samples by selecting those with distinct embeddings.

A model trained with a subset of the full training dataset clearly cannot reach the same level of efficacy as one trained with the entire training dataset. Thus, the optimized subset is only used for initial training, after which fine-tuning is performed with the full training dataset. We show that we are able to achieve a reduction in the required number of training steps, corresponding to 91.36% smaller epochs, on an image classification task, using the CIFAR-10 dataset (Krizhevsky & Hinton, 2009), with minimal sacrifice in the resultant network's accuracy. In such tests, the optimized dataset was just 8.64% the size of the full training dataset. Furthermore, given a fixed time budget, our technique can yield even higher accuracies than a full training approach since epochs under the optimized dataset take less time.

In the next section, we motivate our work with a discussion of the challenges in training neural networks on large datasets. We then provide an overview of a variety of related work in the literature. Following this review, we describe our dataset optimization technique in detail, along with the domain upon which we have evaluated it. A high level, step-by-step overview of the technique is also provided. In subsequent sections, we present our results and a discussion to evaluate this work's impact on the field.

## 2 MOTIVATION

Owing to advances in modern semiconductor manufacturing processes, the industry has seen significant increases in the capacities of modern random access memories. However, as the pace of miniaturization slows and larger machine learning models become the norm, the demands placed upon memories have outstripped their increases in size. For instance, in the 2014 ImageNet Large Scale Visual Recognition Challenge, the image detection dataset required 47GB of storage, while the classification / localization dataset consumed 138GB (Russakovsky et al., 2015). These numbers are only set to grow as researchers tackle larger datasets with richer, higher resolution data.

While these datasets have played a key role in training networks with state-of-the-art accuracy across different tasks, they have also been damaging for training times. The first, less obvious, reason for this is that larger datasets pose significant challenges for existing accelerator architectures (such as the Google Tensor Processing Unit (TPU) (Jouppi et al., 2017), the NVIDIA V100 GPU, and various FPGAs). Unfortunately, most accelerators only have between 4GB and 16GB of memory available on-board, depending on their price and target applications. Since the full dataset cannot fit entirely in accelerator memory, new training examples must be constantly streamed to the accelerators over the PCIe bus, which is approximately two orders of magnitude slower than the on-board memory. Furthermore, if the dataset does not fit in CPU-side memory, transfers may be limited by the computer's underlying storage speed. However, even when CPU-side memory is large enough, training example preprocessing, such as data augmentation and formatting, can overwhelm the CPU if there are not enough cores. From these observations, it can be noted that training optimally on large datasets requires an enterprise-class budget that is out of reach of many researchers.

Additionally, large datasets have revealed an even more relevant problem: training inefficiency due to a lack of scalability. Currently, neural network training consists of the repeated presentation of samples to the model. Unfortunately, while these techniques may be viable for smaller networks or datasets, large datasets have shown that they do not scale well. Despite the wider variety of information present in larger datasets, each example must still be presented to the neural network a roughly similar number of times. This leads to significantly diminishing returns where the added examples increase training time significantly while only increasing accuracy slightly, if at all. If we want to be able to continue to incorporate the ever increasing amount of data available into neural networks, this problem cannot be ignored.

There are a number of existing solutions to both of these challenges, but they fall short. While one could accept that training neural networks on large datasets will be less than ideal, this would constitute giving up and refusing to solve the problem. Instead, we believe that reducing the training set size during only the first part of training will solve these problems with fewer compromises than existing solutions. By reducing the training set size by an order of magnitude, almost all existing datasets can fit entirely in accelerator memory. This requires the data to only be transferred to the accelerator once, eliminating practically all bottlenecks so that the accelerator can proceed at full speed with minimal cost to the rest of the system. Since only fine-tuning must be performed with the full dataset, much of the training time can be accelerated, resulting in significant time savings.

## 3 RELATED WORK

To date, dataset optimization has not been a major topic in deep learning literature. Most studies focus on improving accuracy, and use as much data and training time as necessary. When training and evaluation time is a concern, researchers have started to investigate the use of network architectures that are inherently quicker to train (such as in Yamanaka et al. (2017)).

A key contribution to training time reduction via dataset optimization was promulgated by Dünner et al. (2017). With the goal of allowing datasets to fit entirely in the memory of different compute hardware, a system was developed to intelligently prune datasets through analysis of the duality gap. The authors' approach enables better utilization of hardware in heterogeneous systems. Notably, this work is tailored for support vector machines (SVMs); neural networks and more sophisticated models were out of the work's scope.

While dataset optimization is not a well studied area, embedding spaces are. Embeddings and similar latent representations more frequently form the basis for systems' functionality. One key area is in

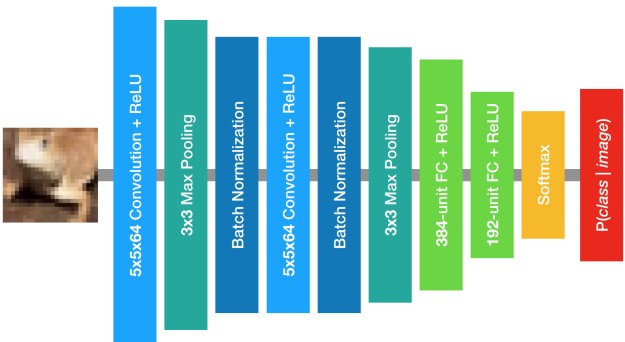

Figure 1: CIFAR-10 classifier model. This standard CNN-based architecture identifies image classes on the CIFAR-10 dataset, serving as a good benchmark for the dataset optimization study in this paper.

generative adversarial networks (GANs) (Goodfellow et al., 2014). GANs learn to generate new data that is similar to a training dataset by jointly training separate generator and discriminator networks. The generator learns to output data, based off of a latent input, that the discriminator deems realistic. These latent input spaces have been found to exhibit unique properties; namely, two latent vectors that are not distant from each other lead to similar outputs when propagated through the generator. This property is a key insight that enables the presented dataset optimization technique as well.

There has also been work in manipulating latent vectors with arithmetic operations. Wu et al. Wu et al. (2016) applied GANs to 3D voxel data to learn dense representations for 3D models. The authors found very interesting behavior when performing addition and subtraction on embeddings. Sequences of operations on embeddings resulted in new, unseen 3D models being generated that follow human intuition. For instance, one can subtract the latent representation of a chair without arms from the representation of a chair with arms to generate the latent representation of the arms feature. This feature representation could be combined with that for an object without arms to generate a version of that object with arms attached. These results demonstrate that latent representations encode high-level data about the objects they represent and that they are more than just a compressed version of the raw input data.

More recent works have aimed to impose desiderata on learned embedding spaces. Hsiao & Grauman (2017) created an embedding for clothing fashion that encodes information for fashion style, while encoding little else. Wang et al. (2016) built a system that learns a joint embedding for images and text, allowing for the construction of systems that can handle images and text in an equal manner.

## 4 APPROACH

To evaluate the technique's efficacy, we have chosen the well-studied task of image classification. Specifically, we are performing classification on the CIFAR-10 dataset (Krizhevsky & Hinton, 2009). We describe our basic classifier model in detail in Subsection 4.1. Following this, we discuss the system's autoencoder architecture and the technique we developed.

### 4.1 CLASSIFIER ARCHITECTURE

Figure 1 illustrates the CIFAR-10 image classifier used in this paper. We follow a basic convolutional neural network (CNN) architecture that takes inspiration from AlexNet (Krizhevsky et al., 2012). The network architecture has been chosen due to its simplicity; achieving state-of-the-art accuracy on CIFAR-10 is not the focus of this paper. $5\times5$ convolutions, $3\times3$ max-pooling, and batch normalization layers arranged in sequence, following typical CNN designs, gradually reducing dimensionality. Notably, inputs to the network are sized $24\times24\times3$, rather than $32\times32\times3$; this enables data augmentation, described in detail in subsection 4.1.2

Since this is a multi-class, single-label classification problem, output logits from the final fully-connected layer are passed through a softmax layer, ensuring that all final outputs correspond to predicted posterior probabilities.

### 4.1.1 TRAINING

The classifier model was trained in an end-to-end fashion with back-propagation, using basic stochastic gradient descent (SGD) with momentum and a learning rate of 0.1.

Training occurs under a standard cross-entropy loss function where $y$ is the network's predicted label and $\bar{y}$ is the ground-truth label:

$$\mathcal{L} = -\frac{1}{n} \sum_{i=0}^{n} \bar{y}_i \log(y_i), \tag{1}$$

where $n$ is the number of elements in both $y$ and $\bar{y}$.

The final fully connected layers of the model have L2 regularization applied during the training process, with a weight decay of 0.004.

To improve the time it takes for the model to converge, an image's mean pixel value is subtracted from every other pixel, both during training and testing. To attain a further speedup, we divide every pixel by each image's variance.

### 4.1.2 DATA AUGMENTATION

Due to the relatively small size of the full CIFAR-10 dataset, the application of data augmentation can be beneficial. Since each CIFAR-10 image is 32×32, and the classifier takes in 24×24 images, there is quite a bit of wiggle room, in the most literal sense, for data augmentation. The network is forced to learn spatial invariance better by randomly selecting 24×24 crops from each full-size image. Each crop is then randomly flipped longitudinally. Following this, images are randomly lightened or darkened and their contrast is randomly perturbed.

### 4.1.3 TESTING

Testing is performed following the standard inference method typically applied to neural networks. Utilizing network checkpoints created while training the network with TensorFlow, the testing network is initialized using a weighted average of the final weights learned during training. To generate the input for testing, the test dataset is loaded and split it into batches that can be passed through the network.

## 4.2 AUTOENCODER ARCHITECTURE

Our technique is flexible with regards to the specific autoencoder design that is chosen. In this paper, we have built a standard hourglass-shaped autoencoder inspired by common designs in the literature. Figure 2 shows the selected autoencoder design in detail. The encoder portion is composed of 3×3 max-pooling layers, batch normalization layers, and basic 5×5 convolutions with ReLU activation functions. The encoder serves to calculate a 216-element embedding for a given input.

The decoder is simple in its design, consisting of three sequential 5×5 deconvolution layers with ReLU activation functions. Deconvolutions are performed as stride-2 transposed convolutions. While transposed convolutions with non-unit strides are susceptible to generating checkerboard artifacts, we have opted to use them over the more robust resize-convolutions (Odena et al., 2016) due to their simplicity. Ultimately the generated reproductions have only instrumental value in the learning of an embedding; any checkerboard artifacts in the reproduced output do not make the learned embeddings any less useful. The decoder's resultant tensor has the same dimensions as the autoencoder's input tensor (i.e., 24×24×3).

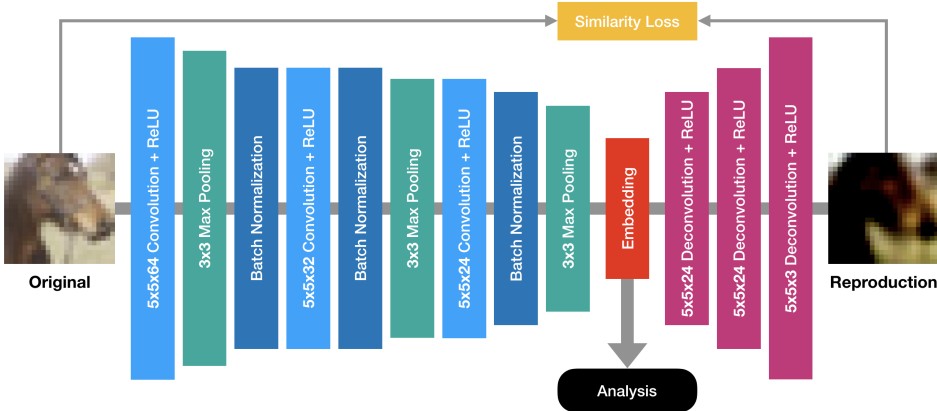

Figure 2: Autoencoder architecture. Data flows from left-to-right through a CNN-based architecture. Resultant embeddings are ultimately analyzed to determine data sampling.

As with the classifier, the autoencoder is trained using SGD with momentum and a learning rate of 0.1. Furthermore, the autoencoder is trained under a basic squared-difference similarity loss:

$$\mathcal{L} = \frac{1}{n} \sum_{i=0}^{n} (y_i - x_i)^2, \tag{2}$$

where $y$ is the generated output, $x$ is the autoencoder's input, and $n$ is the number of elements in both $y$ and $x$. This simple loss function, in essence, forces the network to learn to extract the key features from the input, so that it can reproduce it using said features only. If desired, one could elect to use a more sophisticated loss, such as the Wasserstein distance metric (Gulrajani et al., 2017; Arjovsky et al., 2017), that takes more into account than raw pixel values.

### 4.3 FILTERING EMBEDDINGS

Given a set of embeddings $X$, we would like to find a subset $Y \subset X$, of size $|Y| = k$, where each element in $Y$ is maximally distant from every other element in $X$. That is, $Y$ is the set that maximizes:

$$Y = \operatorname*{argmax}_{Y \subset X} \sum_{i=0}^{k} \sum_{j=0}^{k} \operatorname{dist}(y_i, y_j) \,\Big|\, |Y| = k, \tag{3}$$

where $y_i$ is the $i$th element in $Y$ and $dist$ is a distance metric (e.g., Euclidean distance or cosine distance). Unfortunately, the task of finding such a $Y$ is reducible to the clique problem, which is np-hard. Thus, a different technique or an approximation must be used.

For this work we have chosen to follow Occam's Razor and use a simple, tractable filtering technique. Rather than finding an optimal subset that maximizes distance, we find a subset that maximizes activations for each individual element in the embedding. That is:

$$Y = \bigcup \forall i \in n, \operatorname*{argmax}_{Y' \subset X} \forall y' \in Y', y'_i \,\Big|\, |Y'| = k/n, \tag{4}$$

where $y'_i$ is the $i$th element in the embedding $y'$ and $n$ is the number of dimensions in the embedding space. The argmax in Equation 4 serves to select the top $k/n$ samples that activate element $i$ in the embedding.

For small embeddings and larger datasets with millions of samples, such as ImageNet (Russakovsky et al., 2015), it may be advantageous to also maximize activations for combinations of elements.

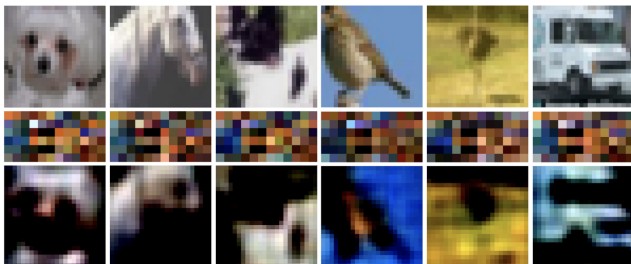

Figure 3: Autoencoder inputs, embeddings, and reproductions (top, middle, and bottom rows, respectively). This figure shows how a basic autoencoder architecture is able to capture and reproduce rich information from images. Notably, reproductions have purely instrumental value; what matters is how well they function as a basis for selecting training examples from the full dataset.

### 4.4 PUTTING IT ALL TOGETHER: OPTIMIZED TRAINING

Our new optimized training methodology expands the standard single-dataset training procedure into the following steps:

1. Fast training: the first portion of training is accomplished quickly with the filtered training dataset.
2. Fine-tuning: the model's accuracy is improved by completing training using the full dataset.

We note that this new training methodology requires no changes to the base model. The only potentially requisite adjustments that could be made are tweaking the learning-rate schedule for each of the two steps to optimize learning. We note that the same learning rates were used across both steps.

These training steps are accompanied by the following one-time per dataset operations:

1. Train the autoencoder: An autoencoder is trained using the full dataset.
2. Evaluate the embeddings: The full dataset is passed through the encoder portion of the autoencoder, resulting in an embedding for each training sample.
3. Analyze the embeddings: A subset of embeddings is found from the full set using the techniques described above.
4. Filter the dataset: Given the samples that are known to correspond to the subset of embeddings, a new, filtered dataset is created. This task happens nearly instantly using a C++ program.

## 5 RESULTS

To validate the presented technique, we chose to apply it to the task of image classification on the popular CIFAR-10 dataset (Krizhevsky & Hinton, 2009). The full classification and autoencoder pipelines were implemented in TensorFlow (Abadi et al., 2015), allowing us to take full advantage of the hardware at our disposal.

All training and evaluation was performed on a single Nvidia Tesla P100 GPU with 16GB of high bandwidth memory (HBM) and NVLink (when using more than one P100). The host for the P100s contained a Xeon E5-2650 v4 and 128GB of RAM. Our baseline consisted of a run with 1024-sample batches for 10,000 steps (i.e., slightly more than 200 epochs), converging on an accuracy of 83.3%. The baseline took 736 seconds to run on the P100, nearly $35\times$ faster than training on an Intel Core i7 CPU.

### 5.1 AUTOENCODER

Qualitatively, the trained autoencoder succeeded in learning an adequate embedding. Figure 3 demonstrates reasonable correspondence between six randomly selected inputs and their reproductions.

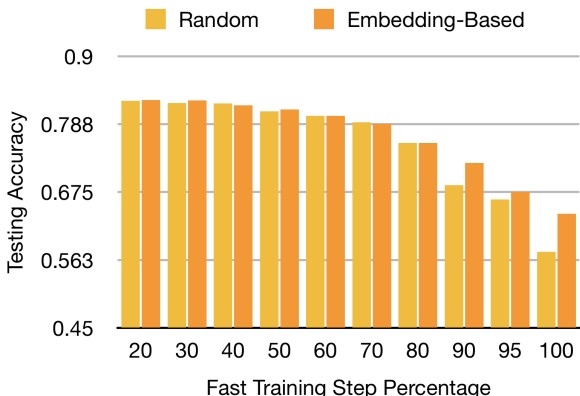

Figure 4: Testing accuracies for varying training step splits and dataset filtering techniques. The merits of an intelligent sampling scheme become clear on training sessions that train on the reduced dataset for a larger percentage of steps.

The autoencoder was trained over 40,000 steps of 1024-sample batches (i.e., slightly more than 800 epochs). The autoencoder's performance in practice is reviewed in subsection 5.2.

In the graphical representation of each embedding in Figure 3, there is an interesting azureous region that recurs on the left side of each image. This phenomenon could be either because the autoencoder is not making full use of the embedding space it has developed, or there are fundamental similarities between the six images that manifest themselves similarly in the embedding space. If the first hypothesis is true, a more sophisticated autoencoder or longer training may be necessary to take full advantage of the available embedding space.

## 5.2 OPTIMIZED TRAINING

In the following experiments, the two aforementioned dataset filtering techniques are compared: random selection and embedding-based filtering. Figure 4 shows how final testing accuracies are impacted by varying fast training splits (i.e., what percent of training steps use the training dataset subset) and different dataset filtering methods. All experiments corresponding to data shown in Figure 4 used a very small optimized dataset that contained just 4,320 samples; that is 8.64% of the full dataset, or only 20 samples per class. To ensure stability and rule out small differences due to different network initializations, presented results are the mean of results from two separate runs.

For most splits, there are only slight differences between the random and embedding-based filtering methodologies, with embedding-based having much less variance between runs than random. However, with a 100 / 0 fast training split (i.e., no training occurs with the full dataset), our embedding-based filtering technique yields an accuracy of 63.9%, while random sampling yields 57.6%. This set of results provides compelling evidence that the filtering technique is effective. Arguably, this large difference only manifests itself at 90 / 10, 95 / 5, and 100 / 0 since fine-tuning is able to smooth over differences in the nonoptimal fast training for the other splits. Furthermore, the small number of classes in CIFAR-10, and the regularity of sample images in the dataset help to alleviate any ill effects caused by random sampling. On more sophisticated datasets, especially those with many classes, the improvement from using embedding-based over random filtering would likely be more prominent.

## 5.3 FIXED TIME BUDGET TRAINING

In many cases, it can be useful for researchers to set an upper bound on training time. The merits of our technique become clear with a fixed time budget. With optimized training, systems can reach a higher testing accuracy than they could otherwise. This happens due to the nature of optimized training, where an individual epoch takes significantly less time. As a result, our optimized training technique allows for a greater number of epochs in a fixed period of time, resulting in better performance.

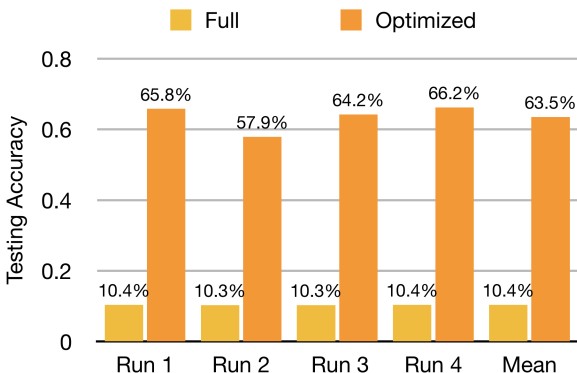

Figure 5: Testing accuracies for fixed time budget training. There is a clear advantage to optimized training due to the larger quantity of epochs that can be presented during training.

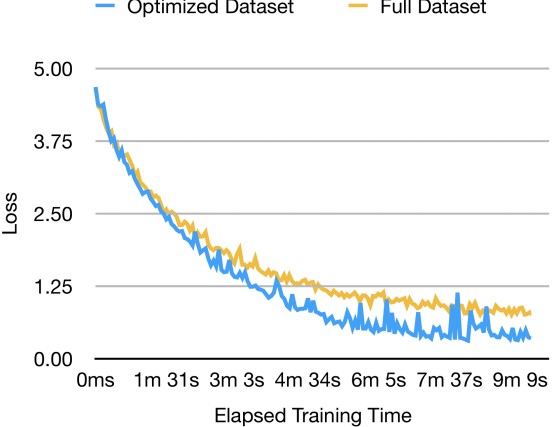

Figure 6: Loss curves for fixed time budget training. The model's loss decreases more rapidly, relative to the elapsed training time, with the reduced dataset than the full dataset.

Figure 5 presents testing accuracies from a series of training sessions that were limited to 7 minutes in length (recall that a full run took approximately 12 minutes). The yellow bars (left) provide results from using the full dataset alone. The orange bars (right) provide results from runs that used 5,000 fast epochs initially (as in the 50 / 50 split shown above), with the remaining epochs using the full dataset. To ensure stability and diminish effects from different network initializations, each experiment was run four separate times, as shown. As in the previous section, an optimized dataset that is 8.64% the size of the full dataset is used.

At the end of the training period the, the models under the optimized training technique had gotten significantly farther along in the training process than those trained with the full dataset, resulting in an average accuracy of 63.5% versus 10.4%. We note that 10.4% is somewhat low for CIFAR-10, this is for a variety of reasons: (1) training purposefully did not reach convergence, (2) the evaluation model is not state-of-the-art, and (3) the task at hand is more challenging since the network uses 24×24 pixel images, rather than the full 32×32 pixel images. We argue that the model's simplicity helps show the technique's merits as a generally applicable technique that is not bound to certain types of architectures.

Figure 6 shows a comparison of loss curves for both full-dataset and optimized training regimes using the same configuration as above (i.e., 8.64% of full dataset). While each individual epoch is not as effective in reducing loss with the optimized dataset, the time each epoch takes is reduced enough to make the optimized training loss to decrease at a faster rate over an elapsed training time. Training with an optimized dataset with ten-times as many samples (i.e., 86.40% of full dataset) resulted

in comparable behavior. We believe this shows the merits of using intelligent dataset optimization during the training process.

# 6 DISCUSSION

The presented results are indicative of our technique's performance on the CIFAR-10 classification task. In sample selection for creating a reduced dataset, the presented embedding-based analysis approach exhibits higher performance than random sampling. In the fixed time budget budget scenario, networks trained with the presented technique result in higher accuracies at the end of each training session than networks trained with the full dataset.

The technique is general and does not depend on any particular kind of task and data. Any tasks with large supervised datasets where embeddings can be formed could in principle benefit from it. Characterizing its extent and applicability is an interesting direction of future work. Another direction is to evaluate other types of autoencoders as part of the system, and analyze how the quality of the embeddings affects the results.

Latent representations have already shown promise in the literature for generating high-quality data in a generative-adversarial learning setting (Goodfellow et al., 2014; Karras et al., 2017). While this paper has focused on reducing datasets, the learned embeddings could be useful in data augmentation as well; for instance, by creating interpolations between real training samples.

Furthermore, we would like to investigate different embedding filtering techniques with more rigor. The $k$-means clustering seems like a reasonable approximation to our initial distance maximization. Principal components analysis (PCA) may be a good precursor to such filtering. The existing technique could also be enhanced by penalizing images that contain too many highly-active features. This modification could improve how fast the network learns by more clearly isolating features to learn early on.

# 7 CONCLUSION

This paper provides a compelling case for the use of dataset optimization to reduce training times, and in improving accuracy with a fixed training time budget. Presented results show this technique's effectiveness on image classification, and with discussed directions for future work, its flexibility will only increase. As a practical method, it should help advance the pace of innovation in machine learning, increasing the tractability of training sophisticated models with large datasets.

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
