# OpenReview forum: "Faster Training by Selecting Samples Using Embeddings"
_ICLR.cc/2019/Conference_

### Official Review · AnonReviewer2 · 2018-11-02
**Training data subset selection to speed up training**

**Rating:** 2
**Confidence:** 5

**Review:**

This paper proposes a way to speed up initial training a model.  The key idea
is to:

1. Train an autoencoder on the full dataset and select a subset of training
examples.  The subset is the union of examples that maximally activate each of
the dimensions of the autoencoder's low-dimensional embedding.

2. Then a target classifier is trained on the subset,

3. followed by final fine-tuning on the full dataset.

The paper is understandably written, although some crucial experimental details
need a bit of guesswork.

Their proposal is evaluated on only one dataset, CIFAR10, using an autoencoder
and classifier of roughly similar design from the initial convolutional layers.

They mention a baseline [classifier training, I presume] classifier training
over ~200 epochs in 736 s (~12 min) to get 83% accuracy.  This skips steps 1.
and 2.  Since this is already fast, CIFAR10 is perhaps too small a dataset
to spur readers to use their proposed method (which does require them to
additionally train an autoencoder) when tackling more ambitious problems.

They do not report the time taken to train their autoencoder for 800 epochs
(step 1.).  For larger networks and images, it might also be important to
investigate whether an autoencoder considerably simpler than the classifier
model can suffice for subset selection; for example, if I want to train a
Resnet-152 classifier can I use a poorer quality autoencoder?  Since
using a randomly selected subset 20% of the original size works about
as well as step 1 for CIFAR10, I cannot judge whether the time taken to
set up and train an autoencoder makes it worthwhile to further reduce
the training subset from 20% to ~8% of the original size.

They do not consider alternative subset selection (1.) methods.  For example,
one might use a pretrained network to select examplar images by a clustering
method (ex. [2]), possibly providing representative images per class.  Other
selection criteria are also possible -- for example, [1] evaluates subset
selection based on "representativeness" vs "diversity" criteria.

They do not compare with many existing approaches to training set compression.
Instead, they dismiss (Sec. 3 "Related Work") most previous work on selecting a
small subset of training examples.  However, googling will quickly find many
papers on subset selection (exactly what they do) as well as related dataset
optimization techniques (such loss-based revisiting of training examplars, or
training example weighting etc.).  For example, review-type article [3]
provides a good introduction to existing subset selection techniques, as well
as references to earlier papers.

It is unclear whether the autoencoder training time is included in their
experiments that fix the total training time to 7 minutes and compare results
with different numbers of fast epochs (step 2.).

No guidelines are given for how to select the dimensionality of the autoencoder
embedding, and how the selection procedure should be done in cases with large
numbers of classes, although they mention the possibility of using combinations
of activations for subset selection.  I do not understand how in problems with
larger numbers of classes I can guarantee that the training subset will contain
at least one representative from each class.  Some alternative subset selection
methods can provide such guarantees, which might be important for training
datasets with class imbalance.

Given that they do not use a very large dataset, where their technique would
really be needed, and that they provide no comparison with other possibly
faster and better ways to select a subset of training examples, I cannot argue
for acceptance of this paper.


[1] "Learning From Less Data: Diversified Subset Selection and
Active Learning in Image Classification Tasks", Kaushal et al.
https://arxiv.org/abs/1805.11191

[2] Li, D., & Simske, S. (2011). Training set compression by incremental
clustering. Journal of pattern recognition research, 1, 56-64.

[3] Borovicka, T., Jirina Jr, M., Kordik, P., & Jirina, M. (2012). Selecting
representative data sets. In Advances in data mining knowledge discovery and
applications. InTech.

---

### Official Review · AnonReviewer1 · 2018-11-02
**Paper on choosing a good subset of data**

**Rating:** 3
**Confidence:** 3

**Review:**

This paper presents the idea of splitting the training process into two phases: fast training on a subset of the original dataset and finetining on the full dataset. To find a good subset of the training dataset it is proposed to train an autoencoder and use its embeddings to choose examples that have large values of the embedding features. The experiments show that on CIFAR-10 dataset this may speed up the convergence.

In general, I like the idea of being smart about which data and in which order to feed to the learner.

Nonetheless, I disagree with several premises of this paper. The paper claims that by making the dataset smaller one can speed up the training by the means of fitting the dataset into the accelerator memory and thus avoiding slow memory copies from CPU to accelerator memory. However, modern deep learning data pipelines are built in a way that has virtually zero overhead, since the data is loaded from disk and preprocessed on CPU and then copied on the accelerator asynchronously (i.e. the GPU doesn’t have to wait for the data, it can process the current batch and at the same time load the next one). Moreover, moving the data to GPU will introduce additional overheads in the case of random data augmentation, since this additional work would have to be done by GPU (while current deep learning frameworks asynchronously do this work on CPU). And finally, the authors claim that most modern datasets can fit to the accelerator memory if reduced 10x, but in my experience the network and it’s activations (which are stored during training) occupies most of the GPU memory even on high end accelerators, not leaving enough space to store a large dataset even after data reduction.
The paper cites Dunner  et al. (2017) as related work that focus on the similar problem: how to find a subset of the dataset to fit it into the GPU memory. However, I would argue that their setup is very different because they are using linear models (such as SVM): their learning steps are very fast compared to CNNs (which makes the memory bandwidth much slower in comparison), they don’t have to store activations of the layers (which allows them to fit much more training samples into GPU memory), and they don’t use data augmentation.

Also, I don’t think that the experimental comparison provides a strong enough evidence supporting the benefits of the proposed scheme. First, the experiments are only done on a small scale dataset (CIFAR-10), which is OK in general, but questionable when the proposed method explicitly targets big data regime and making the training faster. Second, the only baseline considered is choosing subset of data randomly. Third, the optimization method is plain SGD with momentum, while when presenting techniques for faster convergence it would make sense to compare on at least several standard optimization algorithms (e.g. Adam). Finally, the presented results are weak: in Fig 4 any improvements over random baseline are noticeable only after degrading the performance of the network by a large margin (to less than 67% accuracy on CIFAR-10);
Fig 5 looks like it has an error: training on the full dataset performs on the level of random guess after some training, which contradicts the fact that the same network converged to something reasonable in Fig. 6. Also, I believe that the training on the full dataset is strictly better than training on a random subset of data for a few epochs and then finetuning on the full dataset (with the same time budget). The latter sees the same number of updates but with less data, which should only decrease the test performance. If it’s correct, the results in Fig 5 for the full training should look similar (or better) than the results in Fig 4 for the random subset baseline, but it’s very far from being the case.
In Fig 6 I’m not sure what is being compared. Is a train or test loss? If it’s train, then it’s not a fair comparisons, since the two network are optimizing different train losses. I’m also very surprised not to see the moment of training mode transition on the plot (i.e. the moment when the model switched from restricted dataset to the full one), the lack of it can indicate an implementation error.

And finally, I would like to see the text being improved. Right now the language is confusing, for example: “this technique is shown to be effective” (what does it mean “effective” and compared to what?), “Unfortunately, while these techniques may be viable for smaller networks or datasets, large datasets have shown that they do not scale well.” (who have shown that the techniques doesn’t work on large dataset?), “the testing network is initialized using a weighted average of the final weights learned during training” (what does it mean?), “Qualitatively, the trained autoencoder succeeded in learning an adequate embedding.” (what does it mean?), etc.
Also, there is a typo in formulas 1, 2, and 3: it probably should sum up to n-1.
And I didn’t get what formula 4 means, what is “union of (for all i in n)”?
This bit I also didn’t get: “This simple loss function, in essence, forces the network to learn to extract the key features from the input, so that it can reproduce it using said features only. If desired, one could elect to use a more sophisticated loss, such as the Wasserstein distance metric (Gulrajani et al., 2017; Arjovsky et al., 2017), that takes more into account than raw pixel values.”. How can you substitute L2 loss in an autoencoder with Wasserstein metric (which is a metric between probability distributions, not images)?
There is also some missing related work, e.g. the idea mentioned in conclusion on augmenting the dataset in the latent space is presented in DeVries et al. “Dataset augmentation in feature space”.

It would be interesting to connect this work with importance sampling off-policy RL (see e.g. “Prioritized Experience Replay”) and look into sampling dataset points proportional to some importance probability with importance sampling correction.

On the positive side, I really enjoyed the look of the figures and diagrams.

---

### Official Review · AnonReviewer3 · 2018-11-05
**Clear and detailed description of the idea, but weak experimentation**

**Rating:** 3
**Confidence:** 5

**Review:**

Summary:
The manuscript introduces a dataset filtering technique for the purpose of speeding up training of machine learning models.
The technique filters the training set, yielding a subset of examples that are as diverse as possible, according to an autoencoder embedding of the input space. First, one trains a deep autoencoder, whose code layer is used as embedding of the input space. Then, for each element of the embedding the top k training samples are selected which activate that element. This reduced training set is then used for rapid training of the model in the first optimization stage, followed by slower fine-tuning on the complete data set. The experimental section presents a comparison of accuracies after training a simple CNN on CIFAR10 with and without the proposed data filtering under several constraints.

Strengths: The proposed technique addresses the important problem of long training times. The description is very clear and detailed.

Weakness:
My main criticism of this manuscript is that the experimentation is not nearly sufficient to support the central claim that dataset filtering via embeddings, as described in this manuscript, is a “general technique” that “any tasks [...] could in principle benefit from [...]”.
The evidence from the presented experiment is rather weak, as only one architecture and one dataset is selected. Furthermore, there are quite a few confounding factors that I don’t think are compensated by averaging the performances of four training runs. A few recommendations on how to improve the experimental section:
- How are hyperparameters selected? For a fair comparison, separate hyperparameter searches should be performed for training with the full training set and with the filtered set. Simple hyperparameters can influence the performance strongly for a given dataset.
- It requires extensive experimentation to “show the technique’s merits as a generally applicable technique that is not bound to certain types of architectures”, for instance trying different types of architectures and datasets. Of course the technique can be applied to most architectures and datasets. However, the question is whether it often helps, not whether it is technically possible. Does it for example improve convergence speed or performance in a state-of-the-art network trained on ImageNet?
- The heavy use of data augmentation is a confounding factor which adds randomness that is not likely to be compensated by averaging a few training runs. Maybe you could present performances without augmentation.
- You mention momentum is used. For reproducibility, it would be good to state the coefficient used.
- Testing is performed on checkpoints with some form of weighted averaging of final weights. Could you describe the steps in detail for better reproducibility?
- Is the result stable over multiple autoencoder trainings?
- It would be interesting to see the performance before the finetuning stage!

I feel the discussion section could benefit from a few thoughts on the limitations of this approach. For instance, the method might not be the best choice for highly imbalanced classification datasets. Literature on dataset resampling for such scenarios might be worth mentioning in the related work section. Also, the autoencoder’s embeddings are trained to reconstruct the whole image, an objective that gives more importance to patterns that occupy a larger portion of the image. If the downstream task needs attention to detail (e.g. counting of small objects, segmentation in remote sensing or medical imaging, street-number or road-sign detection), the filtering method might also not be much better than random subsampling.

The related work section could also be improved. I see only one work on data set optimization. I’ve seen work using a reducedMNIST dataset, which is probably created by random subsampling, but still more relevant than many of the aspects of embeddings cited in this section (the paragraph about arithmetic operations for instance). Katharopoulos and Fleuret (2018) seems like highly relevant recent work, which should be cited and contrasted against. The evaluation in that work seems very thorough in comparison.

A general recommendation on writing: Try to limit the content to relevant details. For example, a description of hardware specifics (support for NVLink, which is not used) or stating the well-established speed-up when using GPUs for CNNs are not relevant.

Figure 6 could be improved by marking on the time axis, when the fine-tuning sets in.

To summarize my feedback, I like most of the presentation and it is good to see effort towards reducing training times by selecting good training samples, but I think the manuscript requires significant effort to justify acceptance.

---

### Meta-Review · Area_Chair1 · 2018-12-14

**Confidence:** 4
**Recommendation:** Reject

**Metareview:**

The paper proposes a filtering technique to use less training examples in
order to train faster; the filtering step is done with an autoencoder.
Experiments are done on CIFAR-10. Reviewers point to a lack of convincing
experiments, weak evidence, lack of experimental details.
Overall, all reviewers converge to reject this paper, and I agree with them.